# Probiotic Effects of a Marine Purple Non-Sulfur Bacterium, *Rhodovulum sulfidophilum* KKMI01, on Kuruma Shrimp (*Marsupenaeus japonicus*)

**DOI:** 10.3390/microorganisms10020244

**Published:** 2022-01-22

**Authors:** Aoi Koga, Midori Goto, Shuhei Hayashi, Shinjiro Yamamoto, Hitoshi Miyasaka

**Affiliations:** Department of Applied Life Science, Sojo University, 4-22-1 Ikeda, Nishiku, Kumamoto 860-0082, Japan; ak0645akak@gmail.com (A.K.); midorigoto12@gmail.com (M.G.); shayashi@life.sojo-u.ac.jp (S.H.); syamamot@life.sojo-u.ac.jp (S.Y.)

**Keywords:** purple non-sulfur bacteria, *Rhodovulum sulfidophilum*, probiotics, aquaculture, *Marsupenaeus japonicus*, molt, cuticle

## Abstract

Purple non-sulfur bacteria (PNSB) are used as probiotics in shrimp aquaculture; however, no studies have examined the probiotic effects of PNSB in shrimp at the gene expression level. In this study, we examined the effects of a marine PNSB, *Rhodovulum sulfidophilum* KKMI01, on the gene expression of kuruma shrimp (*Marsupenaeus japonicus*). Short-term (3 days) effects of *R. sulfidophilum* KKMI01 on the gene expression in shrimp were examined using small-scale laboratory aquaria experiments, while long-term (145 days) effects of *R. sulfidophilum* KKMI01 on the growth performance and gene expression were examined using 200-ton outdoor aquaria experiments. Gene expression levels were examined using qRT-PCR. Results of the short-term experiments showed the upregulation of several molting-related genes, including cuticle proteins, calcification proteins, and cuticle pigment protein, suggesting that PNSB stimulated the growth of shrimp. The upregulation of several immune genes, such as prophenoloxidase, antimicrobial peptides, and superoxide dismutase, was also observed. In the 145-day outdoor experiments, the average body weight at harvest time, survival rate, and feed conversion ratio were significantly improved in PNSB-treated shrimp, and upregulation of molting and immune-related genes were also observed. When PNSB cells were added to the rearing water, the effective dosage of PNSB was as low as 10^3^ cfu/mL, which was more than a million times dilution of the original PNSB culture (2–3 × 10^9^ cfu/mL), indicating that *R. sulfidophilum* KKMI01 provides a feasible and cost-effective application as a probiotic candidate in shrimp aquaculture.

## 1. Introduction

*Marsupenaeus japonicus* (kuruma shrimp) is a high-value shrimp species, which is mainly cultured in Japan and China. Shrimp diseases caused by fungi, bacteria, and viruses cause serious problems in the shrimp aquaculture industry worldwide [1,2]. The use of probiotics as an eco-friendly and cost-effective approach to raising healthy and pathogen resistant shrimp is increasing [3]. Purple non-sulfur bacteria (PNSB) are characterized by their ability to perform anoxygenic photosynthesis and are widely distributed in soil and water. PNSB are widely applied to agriculture, and the mechanisms on their plant growth-promoting effects have also been intensively studied [4,5], but there are only a limited number of reports on their application in aquaculture [6,7]. In this study, we examined the probiotic effects of a marine PNSB on kuruma shrimp (*M. japonicus*).

Stimulation of the innate immunity of shrimp is one of the most important functions of probiotic microorganisms in shrimp aquaculture [8].

As a crustacean, shrimp lack acquired immunity and rely on innate immunity alone for defense against pathogen infections. The majority of previous studies on the use of probiotic microorganisms, such as *Bacillus* [9,10,11,12,13,14], *Lactobacillus* [15,16,17,18,19], *Lactococcus* [20], *Clostridium* [21,22,23,24,25], *Nocardiopsis* (marine actinomycete) [26], and *Yarrowia* (marine yeast) [27], in shrimp aquaculture have, therefore, focused on the upregulation of various innate immunity genes, such as prophenoloxidase (proPO), superoxide dismutase (SOD), antimicrobial peptides (AMPs), lysozyme, serine proteinase, peroxinectin, lipopolysaccharide and beta-1,3-glucan binding protein (LGBP), and Toll and immune deficiency (IMD) signaling pathways (see Appendix A). Cellular melanotic encapsulation is an effective immune mechanism against pathogens in shrimp, and melanization is activated by the proPO system. Various probiotic microorganisms have been reported to upregulate proPO in shrimp [9,10,11,12,13,15,16,17,19,20,22,24,25,26]. Oxidative burst, the sudden release of ROS to kill microorganisms, is a vital crustacean immune response [28]. Under oxidative burst, shrimp activate ROS-scavenging enzymes, including SOD, to protect themselves from ROS. Several studies observed the upregulation of SOD by probiotic microorganisms in shrimp [9,15,20,21,22,25]. The production of AMPs represents a first-line host defense mechanism of the innate immunity in shrimp. The Toll signaling pathway plays an important role in the activation of AMPs [29,30]. In the Toll pathway, pathogens, such as viruses, bacteria, and fungi, activate the ligand Spatzle that binds to the Toll proteins on the cell membrane to trigger the signaling pathway. The transcription factor Dorsal [31] acts at the final stage and promotes the expression of AMPs, such as anti-lipopolysaccharide factors (ALFs) and crustin [20,29]. Several studies observed the upregulation of Toll and AMPs by probiotic microorganisms in shrimp [9,11,14,18,20,22,24,25,26,27].

Although lactic acid bacteria and *Bacillus* are the major probiotic bacteria used in aquaculture [32], PNSB, such as *Rhodopseudomonas* [33,34] and *Rhodobacter* [35], are also being utilized in shrimp aquaculture. However, the effects of PNSB on gene expression in shrimp and fish have not been reported. In this study, we examined for the first time the effects of PNSB on gene expression in kuruma shrimp. Specifically, we performed short-term (3 days) laboratory aquaria experiments to examine the probiotic effects of *Rhodovulum sulfidophilum* KKMI01, a marine PNSB, on the gene expression in kuruma shrimp, and long-term (145 days) outdoor 200-ton aquaria experiments to investigate the effects of the strain on the growth performance, survival rate, and feed conversion ratio.

## 2. Materials and Methods

### 2.1. Bacterial Strains and Culture Condition

The marine PNSB strain used in this study was *Rhodovulum sulfidophilum* KKMI01 (16S rRNA GenBank/EMBL/DDBJ accession number LC596063) isolated from the seashore of Shimabara Bay (Kami-Amakusa, Kumamoto, Japan) [36]. The PNSB was cultured in glutamate-malate medium [37] containing 3% NaCl under semi-anaerobic conditions in light (light intensity of approximately 100 μM photons/s/m^2^). 

### 2.2. Small-Scale (3 Days) Aquaria Experiments to Examine the Effects of R. sulfidophilum KKMI01 on the Gene Expression in M. japonicus

For the small-scale aquaria experiments in the laboratory, 50 healthy shrimp with an average body weight of approximately 1.5 g were obtained from a commercial shrimp farm (Ebi-no-Miyagawa Co., Ltd., Amakusa, Kmamoto, Japan). The shrimp were fed a basal diet and acclimatized for two days before the start of the experiment, and then randomly assigned to six groups. For the PNSB treatment experiments, 10 L volume of plastic boxes (length 35 cm × width 22 cm × height 13 cm) were used, and 3 L of natural seawater (pH 8.0–8.1, 28 ppt salinity; Naseem 800, Japan QCE bluelab, Shizuoka, Japan) was added to each box. Five shrimp were kept in one aquarium and the rearing water was changed every day. The water temperature was maintained at 24–26 °C, and water aeration was performed continuously using a small air pump for aquaria (e-AIR 2000SB, GEX Co., Ltd., Higashiosaka, Japan). For the treatment of shrimp by *R. sulfidophilum* KKMI01, two methods were examined. One method was to feed the shrimp with the feed containing live (10^6^ colony forming unit (cfu)/g) or dead (10^6^ equivalent cfu/g) *R. sulfidophilum* KKMI01 cells. The other method was to add the live cells directly into the rearing water at concentrations of 10^1^ cfu/mL, 10^3^ cfu/mL, or 10^5^ cfu/mL. To prepare dead (killed) PNSB cells, the cells were collected via centrifugation at 5000× *g* for 10 min, and the cell pellet was dried at 60 °C for 6 h. The dried cell pellet was resuspended in distilled water and disrupted via sonication. To prepare the feed containing live or dead cells (10^6^ equivalent cfu/g), 1 mL of the cell suspension (concentration: 10^7^ equivalent cfu/mL) was dropped slightly to 10 g of dry feed in a 50 mL tube, and mixed by shaking. These PNSB treatment conditions are abbreviated in the text as LF (live cells in feed), DF (dead cells in feed), W1 (10^1^ cfu/mL in water), W3 (10^3^ cfu/mL in water), and W5 (10^5^ cfu/mL in water).

### 2.3. Long-Term (145 Days) 200-Ton Aquaria Experiments

Two outdoor 200-ton aquaria located in Uruma, Okinawa, Japan, were used for the experiments. The aquaria were filled with 150 tons of sand-filtered natural seawater, and 1000 shrimp post-larvae (PL, 16 d, approximate average body weight of 0.15 g) were released into the water. Aeration of the water was performed continuously using a pipeline on the bottom of the aquaria. For the PNSB treatment, *R. sulfidophilum* KKMI01 cells were added to water at a concentration of 10^3^ cfu/mL. The temperature, pH, and dissolved oxygen (DO) levels were recorded daily at 08:00 a.m. and 03:00 p.m. The range of pH during the experiment was pH 7.7–8.3. The water temperature at 03:00 p.m. during the experimental period is shown in the Appendix A. No water change was performed during the first 30 experimental days, after which approximately 10% of the water was exchanged with fresh seawater once a week, and PNSB cells for the exchanged fresh seawater were added at a concentration of 10^3^ cfu/mL. The amount of feed per day was approximately 5% of the shrimp body weight. The control and PSNB-treated shrimp were fed with the same amounts of feed during the first 26 experimental days, after which feed leftovers were checked daily, and the amount was adjusted based on the leftovers.

On day 70 after the start of the experiments, 20 and 30 shrimp from the control and PNSB-treated groups, respectively, were collected to measure the body weight, and on day 145 all shrimp were harvested and the body weights of individual shrimp were measured. Both on day 70 and day 145, RNA samples were isolated from 3 shrimp of each group.

### 2.4. qRT-PCR

For qRT-PCR, RNA samples were isolated from the body parts of shrimp, including the hepatopancreas and intestine, using Isogen (Nippon Gene Co., Ltd., Tokyo, Japan). Briefly, the samples were homogenized using a homogenizer (Physcotron, Microtech Co., Ltd., Funabashi, Japan). The RNA samples were then purified using the RNeasy kit (Qiagen, Venlo, The Netherlands) and treated with RNase-free DNase (Qiagen). The cDNA was synthesized using the PrimeScript II 1st strand cDNA Synthesis Kit (Takara Bio Inc., Tokyo, Japan) with a random hexamer primer. The MyGo Pro Real-time PCR System (IT-IS Life Science Ltd., Dublin, Ireland) was used with a MyGo Green Mix Universal ROX kit. The nucleotide sequences of the primers used for qRT-PCR are shown in the Appendix A. The qRT-PCR was performed under the following conditions: a temperature of 94 °C for 2 min, 40 cycles of 94 °C for 10 s, and 60 °C for 30 s, and a final dissociation protocol from 65 °C to 95 °C. The delta–delta Ct method [38] was used to calculate the relative mRNA expression level with the eukaryotic elongation factor 1-alpha (GenBank/EMBL/DDBJ accession no. AB458256) used as the reference gene. Two technical replicate qRT-PCR reactions were run for each biological replicate.

### 2.5. Statistical Analysis

For the qRT-PCR analyses, the difference was tested using one-way ANOVA, post hoc Duncan multiple range tests for short-term experiments, and Student’s *t*-test for pairwise comparison in long-term experiments. The differences in average body weight and survival rates in the outdoor 200-ton aquaria experiments were examined using Student’s *t*-test and Fisher’s exact test, respectively. All data are expressed as the mean ± standard deviation (S.D.). 

## 3. Results

### 3.1. Genes Upregulated by PNSB Treatment in Short-Term Experiments

The effects of *R. sulfidophilum* KKMI01, a marine PNSB, on gene expression in *M. japonicus* were examined using small-scale laboratory aquaria experiments. The shrimp were treated with PNSB for 3 days. For the PNSB treatment, two methods (five conditions) were examined as described in Section 2, Materials and Methods. After 3 days, RNA was extracted from the body parts, including the hepatopancreas and intestine, and gene expression was examined by qRT-PCR.

#### 3.1.1. Upregulation of the Genes Related to Molting by PNSB

Crustaceans, including shrimp, have a rigid exoskeleton composed of a layered cuticle that covers the soft body parts. This means that growth only occurs at each molt, accompanied by the synthesis of a new cuticle. The cuticle consists of CaCO_3_ salts and organic substances, such as chitin, proteins, lipids, and glycosaminoglycans [39]. In the present study, we observed a high upregulation of several genes related to cuticle synthesis following PNSB treatment (Figure 1 and Figure 2).

DD1, DD5, DD9A, and DD9B are cuticle proteins [40,41,42] that bind to chitin to form a chitin/protein complex. These cuticle proteins were highly upregulated in all five PNSB treatment conditions (Figure 1), and especially DD5 was upregulated more than a hundred times. Among all the genes of cuticle proteins examined, the effect of PNSB was most prominent in the W3 condition.

Figure 2 shows the effects of PNSB on the gene expression of other cuticle synthesis-related proteins. CaCO_3_ adds rigidity to the cuticle, while the crustocalcin protein that has a Ca^2+^-binding site plays an important role in the calcification of the exoskeleton [43]. In this study, crustocalcin was upregulated in LF, DF, W3, and W5 conditions. In addition, the sarco/endoplasmic reticulum Ca^2+^-ATPase was upregulated by several folds, for instance 3-fold in the W3 condition, in all five experimental conditions. This protein plays an important role in calcium metabolism during the molting process [44]. Carotenoids are commonly used for external coloration by crustaceans, and crustacyanin is a carotenoprotein complex found in the exoskeleton of crustaceans [45]. The crustacyanin gene was upregulated in all PNSB treatment conditions.

The upregulation of the genes related to cuticle synthesis suggested the growth-promoting effects of PNSB; therefore, the upregulation of muscle myosin genes by PNSB was expected. In this study, we examined the effects of PNSB on the expression of myosin heavy chain b (MYHb), which is one of the most highly expressed skeletal muscle myosin [46], and we observed a two to eight-fold upregulation of MYHb in all five conditions (Figure 2). This result supports the growth-promoting effect of PNSB. Among all the genes examined, the effect of PNSB was most prominent in the W3 condition. 

#### 3.1.2. Upregulation of the Innate Immunity Genes by PNSB Treatment

The majority of previous studies on the use of probiotic microorganisms in shrimp aquaculture have focused on the upregulation of various innate immunity genes (see Appendix A). In this study, we also observed a significant upregulation of proPO by PNSB in LF, DF, W3, and W5 conditions (Figure 3). The upregulation of SOD2 was also observed in LF, W1, W3, and W5 conditions (Figure 3), indicating that oxidative burst was induced by PNSB, similar to other probiotic microorganisms. 

We also observed the upregulation of several other genes related to shrimp innate immunity, including heat shock protein 70 (HPS70), serine proteinase inhibitor (Serpin), and diphthamide biosynthesis protein 7 (Dph7) (Figure 3). HSP70 acts as a molecular chaperone and stress protein, but it also plays an important role in innate immune responses. In *Litopenaeus vannamei* (Pacific white shrimp), the injection of recombinant HSP70 protein induced the upregulation of several immune-related genes involved in the NF-κB signal transduction pathway, AMP synthesis, and ProPO activating system, and the shrimp receiving injection showed higher resistance to pathogenic *Vibrio parahaemolyticus* [47]. Serpin expression in *M. japonicus* (MjSerp1) was upregulated by *Vibrio anguillarum* infection and inhibited microbial serine protease [48]. The injection of recombinant MjSerp1 protein to shrimp suppressed *V. anguillarum* infection, and MjSerp1 knockdown using RNA interference impaired the resistance to *Vibrio* in shrimp. Meanwhile, Dph7 is a vital protein for diphthamide biosynthesis in archaea and eukaryotes, and expression of Dph7 was reportedly induced by white spot syndrome virus (WSSV) infection [49]. Dph7 knockdown in *M. japonicus* resulted in the downregulation of various immune factors, such as Toll, ProPO, p53, tumor necrosis factor-α, and signal transducer and activator of transcription, suggesting that Dph7 plays a regulatory role in the immunological reaction of shrimp. In this study, we observed significant upregulation of HSP70 and Dph7 in W3, and of Serpin in W1, W3, and W5 conditions.

Figure 4 shows the effects of PNSB on the expression of several Toll pathway-related genes. Although there was no statistical significance due to the large variation in qRT-PCR data, we observed upregulation of Toll 1, Toll 2, Dorsal, two ALFs (ALF and ALF D), and crustin-like genes by PNSB. In addition, the IMD pathway is an important signaling cascade that regulates the expression of AMPs [30]. We observed an approximately two-fold upregulation of IMD in the W3 group (Figure 4).

The effects of live or dead PNSB cells were not significantly different on the cuticle and immune gene expression in shrimp, indicating that the effects of PNSB might be attributed to some cellular components. When the PNSB cells were added directly into the rearing water, the W3 condition seemed to be the most appropriate condition, and the W1 and W5 conditions were under- and over-dosages, respectively.

Although we do not have sufficient data, we conducted some challenge tests with shrimp using a shrimp pathogenic bacterium, *Vibrio alginolyticus*, and an enhanced resistance was observed by PNSB treatment. A preliminary result of the challenge test is shown in the Appendix A.

Based on the results in the short-term experiments, we chose the W3 condition as the most appropriate for application and subsequently examined the effects of *R. sulfidophilum* KKMI01 on the growth performance, survival rate, and feed conversion ratio in kuruma shrimp aquaculture using long-term outdoor experiments on 200-ton aquaria.

### 3.2. Outdoor 200-Ton Aquaria Experiments

The results of the short-term (3 days) PNSB treatments indicated the upregulation of genes related to cuticle biosynthesis, suggesting that the growth of shrimp was promoted by PNSB treatment. To examine the effects of *R. sulfidophilum* KKMI01 on the growth performance, survival rate, and feed conversion ratio in kuruma shrimp aquaculture, a long-term (145 days) experiment was conducted using two 200-ton outdoor aquaria, from 15 October 2018, to 9 March 2019, in Uruma City, Okinawa, Japan. Considering the results of the short-term experiment (Figure 1, Figure 2, Figure 3 and Figure 4), the *R. sulfidophilum* KKMI01 cells were added to the water at a concentration of 10^3^ cfu/mL for the PNSB treatment. The control and PSNB-treated shrimp were fed with the same amounts of feed (approximately 5% of body weight) during the first 26 experimental days, after which the feed leftovers were checked daily, and the amount was adjusted based on the leftovers. The daily amounts of feed for the control and PSNB groups are shown in the Appendix A. 

On day 70 after the start of the experiments, 20 and 30 shrimp from the control and PNSB-treated groups, respectively, were collected to measure the body weight and isolate RNA samples. Figure 5a,b show the average body weight of each group and a photograph of the sampled shrimps. We observed a significant increase (*p* < 0.001) in the average body weight of the PNSB group compared with that in the control group. 

Expression of the cuticle and immune proteins, examined in the short-term experiments (Figure 1, Figure 2, Figure 3 and Figure 4), was examined using qRT-PCR (Figure 6 and Figure 7). In the cuticle proteins, DD1, DD5, and DD9A were significantly and highly upregulated, while DD9B was slightly downregulated (Figure 6). In contrast, there was no change in the expression levels of crustocalcin, Ca-ATPase, crustacyanin, and MYHb (Figure 7). 

In immune proteins, we observed significant upregulation of SOD2 and ALF, while the other proteins showed no significant upregulation (Figure 8a,b).

At 145 days after the start of the experiments, all shrimp were harvested and the body weights of individual shrimp were measured. Table 1 shows the average body weight, survival rate, total body weight, total feed amount, and feed conversion efficiency (FCE) of control and PNSB treated shrimp. In the PNSB treated group, a significant increase (1.54-fold) was observed in the average body weight and survival rate (8.3 point). The total body weight increased by 1.76-fold, and the FCE also improved by approximately 10% (1.58/1.76) in the PNSB treated group. 

We also partially examined the gene expression levels of some cuticle and immune proteins of shrimp sampled on day 145, but the difference between the PNSB-treated and control groups was smaller than that of shrimp sampled on day 70.

## 4. Discussion

In this study, we examined the effects of a marine PNSB, *R. sulfidophilum* KKMI01, on the gene expression in kuruma shrimp (*M. japonicus*). The effects of lactic acid bacteria, *Bacillus*, and *Clostridium* on the gene expression levels in shrimp have been well studied (see Appendix A). To the best of our knowledge, there have been no studies on the effects of PNSB on the gene expression in shrimp, despite the fact that PNSB has been applied in shrimp aquaculture [33,34,35]. 

The most interesting and novel finding of the present study was the upregulation of several genes related to cuticle synthesis by PSNB (Figure 1 and Figure 2). This finding suggests that PNSB stimulates the growth of shrimp, as shrimp have a rigid exoskeleton made of cuticle, and molting is necessary for shrimp growth. Santos et al. [50] compared the RNA-seq data between shrimp groups with higher weight averages (higher growth) and those with lower weight averages (lower growth), and identified several key genes involved in weight gain. They found that cuticle-related genes displayed significantly higher expression levels in higher growth shrimp, suggesting their roles in shrimp growth and weight gain. In the present study, the growth-stimulating effects of PNSB were observed by the high upregulation of cuticle proteins and muscle myosin (Figure 2). 

Based on the results of the short-term laboratory aquaria experiments, we then examined the effects of PNSB on the growth performance of shrimp using outdoor 200-ton aquaria experiments. We found that the growth of shrimp was significantly increased, and the upregulation of the expression of some cuticle proteins was also observed (Figure 5 and Figure 6). According to a meta-analysis by Toledoa et al. [51], there have been 60 studies, from 1980 to 2017, on the effects of probiotics on the growth performance of shrimp. Several studies reported the upregulation of proteins related to innate immunity, such as SOD [52], proPO [53], and lysozyme [54]; however, to the best of our knowledge, the present study is the first to report the upregulation of cuticle proteins by probiotic microorganisms, indicating the direct growth-stimulating effects of PNSB. 

In both short-term (3 days) and long-term (145 days) experiments, we observed an obvious upregulation in the expression of cuticle-related proteins. However, the qRT-PCR data showed extremely wide variations. For instance, in the W3 condition of the short-term experiment (Figure 1), the relative expression levels of DD5 in four individual shrimp samples were 1.8-, 28-, 278-, and 5582-folds higher than that of the control shrimp, resulting in an average value of 1472-fold with a standard deviation of 2742. This extremely wide variation may be due to the molting-stage-dependent expression pattern of cuticle proteins. Molting of shrimp is classified into three main stages: postmolt, intermolt, and premolt, and cuticle proteins are highly specific to certain molting stages [40,41,42]. In our study, the individual shrimp used for qRT-PCR analyses were randomly taken from various stages of the molting cycle. Thus, future studies should consider the molting cycle of the shrimp to understand the exact effects of PNSB on the expression of these proteins. 

Shrimp do not have acquired immunity; therefore, innate immunity plays a crucial role in defense against pathogens. In previous studies on various probiotic microorganisms (see Appendix A), the upregulation of proteins related to innate immunity, such as proPO, SOD, AMPs, lysozyme, serine proteinase, Toll, IMD, and HPS70, has been reported. PNSB are applied in the shrimp aquaculture in some Asian countries [33,34,35], and the present study observed for the first time the upregulation of immune-related proteins by PNSB, providing supportive evidence for the effectiveness of PNSB as a probiotic in shrimp aquaculture. 

However, we observed some inconsistencies in gene expression patterns between the results of the short-term and long-term experiments. For example, crustacyanin was significantly upregulated in the short-term experiment (Figure 2), whereas it was downregulated in the long-term experiment (Figure 7). A possible explanation for this inconsistency is that the PNSB had a stimulatory effect on gene expression, and long-term exposure to PNSB caused acclimation. 

We examined two methods for PNSB treatment: adding PNSB (live or dead) to the feed and adding live PNSB to the rearing water. The dosage in feed (10^6^ cfu/g feed) in this study was comparable with those (10^3^ to 10^8^ cfu/g feed) in the previous studies (see Appendix A). However, considering the low effective dosage (10^3^ cfu/mL) in the method of adding cells to rearing water, the 10^6^ cfu/g feed dosage seemed to be an over dosage, especially for the dead cells. Thus, the effects of PNSB at a lower dosage in feed should be examined in the future. In the case of adding probiotics to the rearing water, previous studies have used 10^6^ cfu/mL of *Ectothiorhodospira shaposhnikovii*, a photosynthetic purple sulfur bacterium [55]; 10^6^ cfu/mL of *Bacillus licheniformis* and *B. subtilis* [56]; and 10^5^ to 10^6^ cfu/mL of *Bacillus subtilis* E20 [57]. In the present study, we examined 10^1^, 10^3^, and 10^5^ cfu/mL of PNSB and found that 10^3^ cfu/mL was the most appropriate dosage. The 10^1^ and 10^5^ cfu/mL of PNSB seemed to be under- and over-dosages, respectively. 

We also examined the method of adding dead cells into the rearing water at the concentration of (10^3^ cfu/mL equivalent), and found that this method worked as well as adding live cells (data not shown), indicating that some cellular components were responsible for the effect. The cell density of a batch culture of *R. sulfidophilum* KKMI01 in the stationary phase is about 10^9^ cfu/mL and 5 mg cell dry weight (d.w.)/mL. The dosage of W1 (10^1^ cfu/mL), W3 (10^3^ cfu/mL), and W5 (10^5^ cfu/mL) can, therefore, be calculated to be 0.05, 5, and 500 ng cell d.w./mL, respectively. Based on this calculation, the active principle in PNSB cells, which stimulated the immunity of kuruma shrimp, was supposed to show its bioactivity at the concentration of as low as pg/mL to ng/mL order. We expect lipopolysaccharide (LPS) of PNSB as a possible candidate of this active principle because generally, LPS exhibits its bioactivity at this order [58]. In addition, Takahashi et al. reported that LPS of *Pantoea agglomerans*, a Gram-negative bacterium that grows symbiotically with various plants, enhanced the phagocytic activity and phenoloxidase (PO) activity in *M. japonicus*, and also enhanced their resistance to penaeid rod-shaped DNA virus (PRDV) [59]. LPS is also known as endotoxin because of its ability to induce toxic inflammatory responses. The over-dosage effect of PNSB cells observed in W5 condition may possibly be explained by the deleterious effects of LPS. The detailed study on the effects of LPS from *R. sulfidophilum* KKMI01 in *M. japonicus* are under progress in our laboratory.

The much lower effective dosage (10^3^ cfu/mL) for *R. sulfidophilum* KKMI01 compared with those in the previous studies indicated the superior probiotic effects of this PNSB strain, suggesting a more feasible application in the water of open shrimp ponds. Specifically, since the concentration of PNSB in the stationary phase of batch culture is usually at 10^9^ cfu/mL, the amounts of culture required to obtain the concentrations of 10^1^ cfu/mL, 10^3^ cfu/mL, and 10^5^ cfu/mL in 200-ton aquaria are 2 mL, 200 mL, and 20 L, respectively. The amounts of culture to obtain the same PNSB concentrations for 1 ha open shrimp pond (water depth 2 m, volume of water 20,000 tons) are calculated to be 200 mL, 20 L, and 20,000 L. Therefore, the concentration of probiotic bacteria in the previous studies (10^5^ to 10^6^ cfu/mL) can be applied only to a hundred tons of aquaria, but the effective concentration (10^3^ cfu/mL) of PNSB found in this study highlights its practical application even to open ponds. 

PNSB, such as *Rhodopseudomonas* [33,34] and *Rhodobacter* [35], have been used for shrimp aquaculture in some Asian countries. These PNSB are freshwater strains and marine PNSB strains are expected to show better performance in marine aquaculture. Marine strains of *Bacillus* [9], lactic acid bacteria [19,20], actinomycetes (*Nocardiopsis alba*) [26], and yeast [14,27] have been reported as promising probiotic candidates in marine aquaculture. In our previous study [36], we reported that *R. sulfidophilum*, a marine PNSB, commonly inhabits the intestinal tract of some marine fish. Therefore, *R. sulfidophilum* is a good probiotic candidate for marine aquaculture. Among the many *R. sulfidophilum* strains examined previously [36], *R. sulfidophilum* KKMI01 that is used in the present study was the most closely related to *R. sulfidophilum* strains inhabiting the fish intestinal tract, based on the nucleotide sequence of the *pufC* gene, which codes for the photosynthetic reaction-center-bound cytochrome subunit. Overall, these findings highlight that *R. sulfidophilum* KKMI01 is a promising candidate for probiotics in aquaculture. 

In conclusion, marine *R. sulfidophilum* KKMI01 stimulated the growth and upregulated the immunity of shrimp. The effective dosage in rearing water was as low as 10^3^ cfu/mL, which was more than a million times dilution of the original PNSB culture (10^9^ cfu/mL). Thus, *R. sulfidophilum* KKMI01 provides a feasible and cost-effective application as a probiotic candidate in shrimp aquaculture. 

## Figures and Tables

**Figure 1 microorganisms-10-00244-f001:**
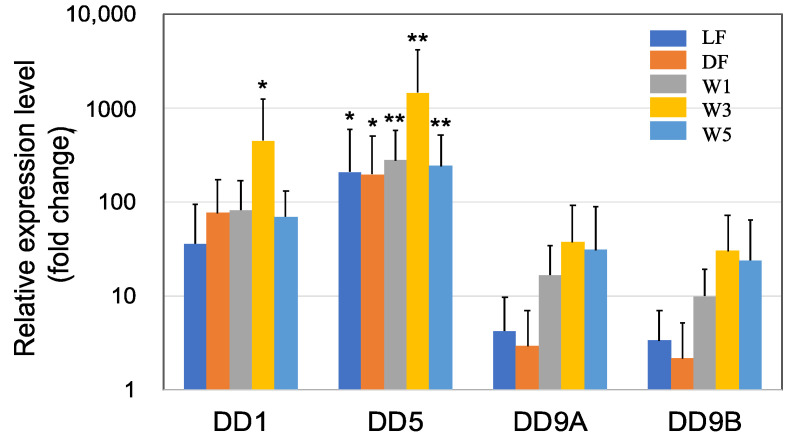
Quantitative real-time PCR (qRT-PCR) analysis of the expression of cuticle proteins DD1, DD5, DD9A, and DD9B in the short-term (3 days) experiments. The shrimp were treated with PNSB for three days, and the expression levels of the genes were evaluated using qRT-PCR. The five PNSB treatment conditions included feeding shrimp containing live or dead (killed) PNSB cells at the concentration of 10^6^ cfu/g feed, and adding live PNSB cells into the rearing water at concentrations of 10^1^, 10^3^, or 10^5^ cfu/mL. Each bar represents the mean fold change relative to the control ± S.D. of four independent biological replicates. Asterisk indicates significant difference with control at *p* < 0.05 (*) and *p* < 0.01 (**).

**Figure 2 microorganisms-10-00244-f002:**
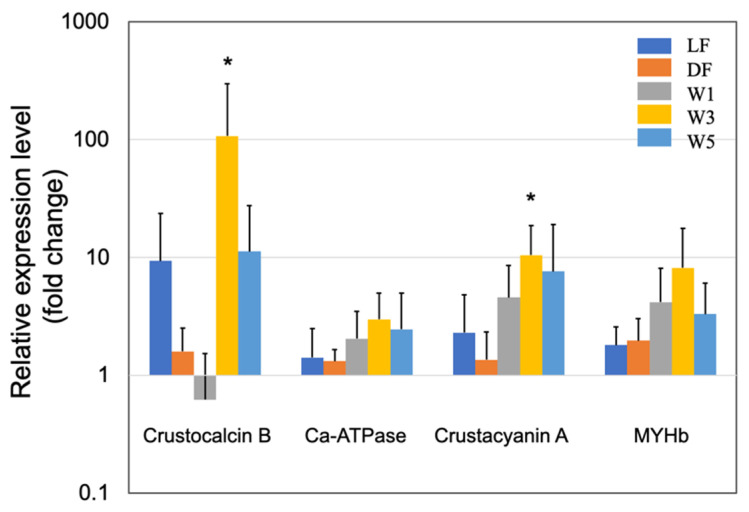
Quantitative real-time PCR analysis of the expression of crustocalcin B, sarco/endoplasmic reticulum Ca^2+^-ATPase, crustacyanin A, and myosin heavy chain b (MYHb) in the short-term (3 days) experiments. The experimental conditions were the same as in Figure 1. Each bar represents the mean fold change relative to the control ± S.D. of four independent biological replicates. Asterisk indicates significant difference with control at *p* < 0.05 (*).

**Figure 3 microorganisms-10-00244-f003:**
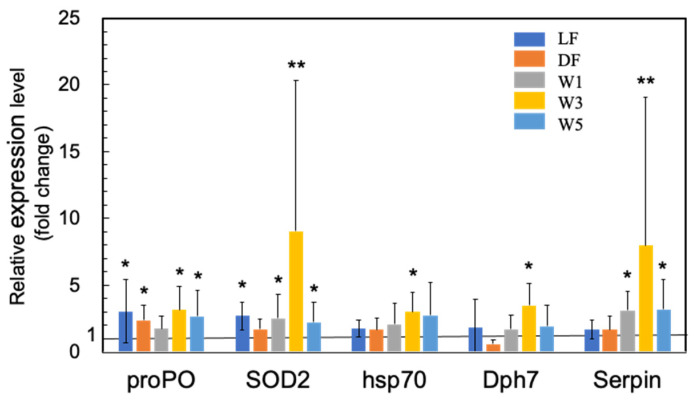
Quantitative real-time PCR analysis of the expression of proPO, SOD2, HPS70, Dph7, and Serpin in the short-term (3 days) experiments. The experimental conditions were the same as in Figure 1. Each bar represents the mean fold change relative to the control ± S.D. of four independent biological replicates. Asterisk indicates significant difference with control *p* < 0.05 (*) and *p* < 0.01 (**).

**Figure 4 microorganisms-10-00244-f004:**
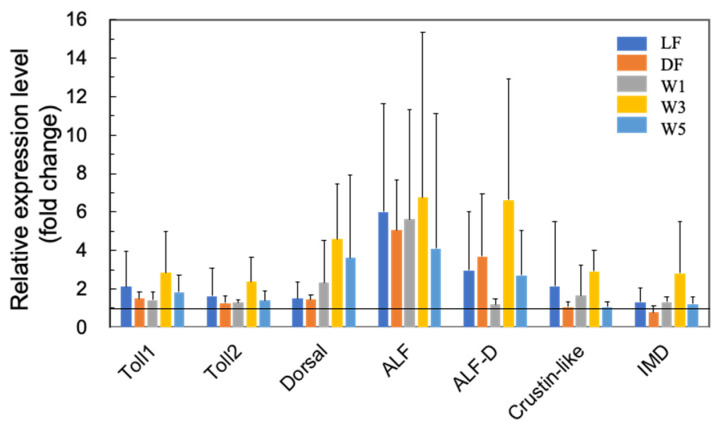
Quantitative real-time PCR analysis of the expression of Toll-pathway-related proteins, including Toll 1, Toll 2, Dorsal, two ALFs (ALF and ALF D), and crustin-like genes, in the short-term (3 days) experiments. The experimental conditions were the same as in Figure 1. Each bar represents the mean fold change relative to the control ± S.D. of four independent biological replicates.

**Figure 5 microorganisms-10-00244-f005:**
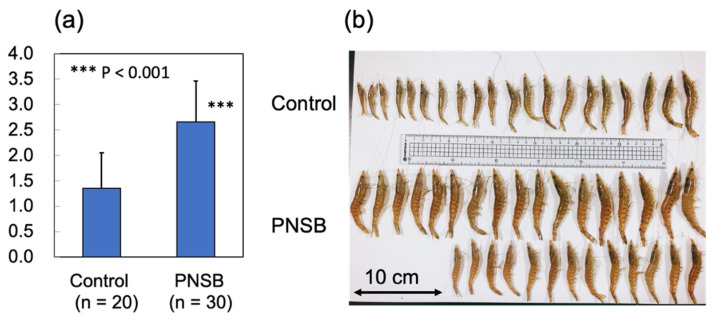
Effect of *R. sulfidophilum* KKMI01 on the growth of kuruma shrimp (*M. japonicus*) in long-term outdoor 200-ton aquaria experiment. (**a**) Average body weights of control (*n* = 20) and PNSB-treated (*n* = 30) shrimp 70 days after the start of the experiments. Values represent means ± S.D. Asterisk (***) indicates significant difference with control at *p* < 0.001. (**b**) Photograph of control and PNSB-treated shrimp.

**Figure 6 microorganisms-10-00244-f006:**
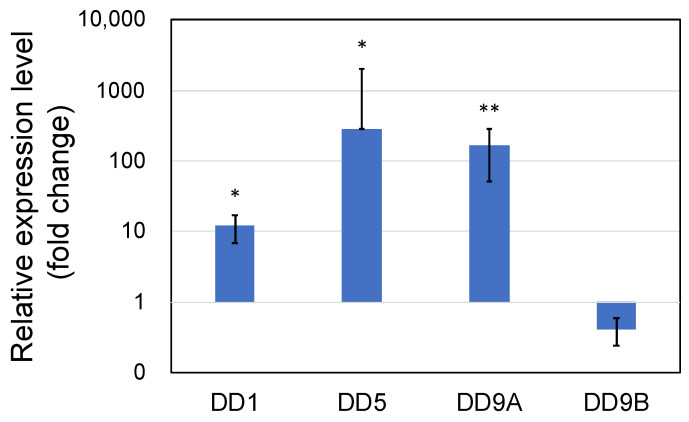
Quantitative real-time PCR analysis of the expression of cuticle proteins DD1, DD5, DD9A, and DD9B in the long-term (70 days) outdoor 200-ton aquaria experiments. The shrimp were treated with PNSB for 70 days in the 200-ton aquaria, and expression levels of the genes were evaluated using qRT-PCR. Live PNSB cells (10^3^ cfu/mL) were added into the rearing water. Each bar represents the mean fold change relative to the control ± S.D. of three independent biological replicates. Asterisk indicates significant difference with control at *p* < 0.05 (*) and *p* < 0.01 (**).

**Figure 7 microorganisms-10-00244-f007:**
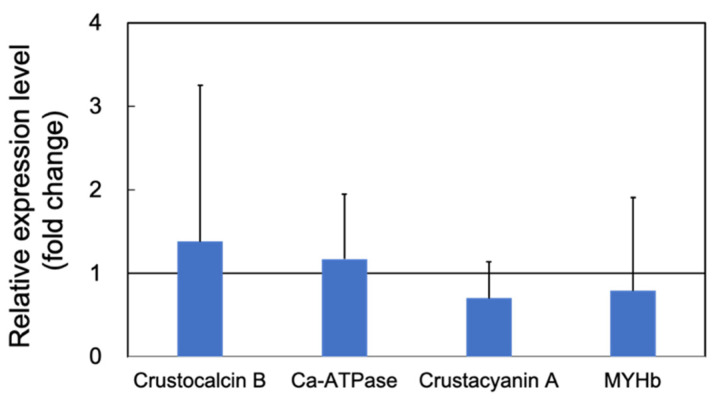
Quantitative real-time PCR analysis of the expression of crustocalcin B, sarco/endoplasmic reticulum Ca^2+^-ATPase, crustacyanin A, and myosin heavy chain b (MYHb) in the long-term (70 days) outdoor 200-ton aquaria experiments. The experimental conditions were the same as in Figure 6. Each bar represents the mean fold change relative to the control ± S.D. of three independent biological replicates.

**Figure 8 microorganisms-10-00244-f008:**
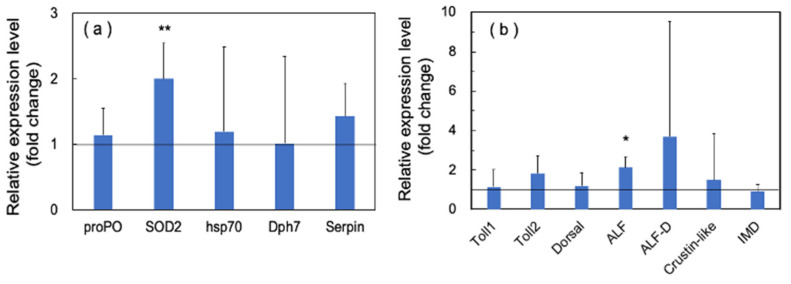
Quantitative real-time PCR analysis of the expression of immune proteins in the long-term (70 days) outdoor 200-ton aquaria experiments. The experimental conditions were the same as in Figure 6. Each bar represents the mean fold change relative to the control ± S.D. of three independent biological replicates. (**a**) Effect of PSNB on the expression of proPO, SOD, HPS70, Dph7, and Serpin genes. (**b**) Effect of PSNB on the expression of Toll-pathway-related genes, including Toll 1, Toll 2, Dorsal, two ALFs (ALF and ALF D), and crustin-like genes. Asterisk indicates significant difference with control at *p* < 0.05 (*) and *p* < 0.01 (**).

**Table 1 microorganisms-10-00244-t001:** Average body weights, number of survivors, survival rates, total harvested body weights, total feed amounts, and feed conversion efficiencies (FCE) in the control and *R. sulfidophilum* KKMI01-treated shrimp.

	Average Body Weight (g)	Final Number of Shrimp ^1^	Survival Rate (%)	Total Body Weight (g)	Total Feed Amount (kg)	FCE ^2^
Control	7.73	589	58.9	4552	7995	1.76
*R. sulfidophilum*-fed	11.9 ***	672 ***	67.2	8024	12,650	1.58

^1^ Initial shrimp number = 1000. ^2^ FCE (feed conversion efficiency) = (total feed amount)/(harvested total body weight). *** *p* < 0.001 (Student’s *t*-test for average body weight, and Fisher’s exact test for final number of shrimp).

## Data Availability

The data presented in this study are available upon request from the corresponding author.

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
