# Peer review of "Probiotic Effects of a Marine Purple Non-Sulfur Bacterium, Rhodovulum sulfidophilum KKMI01, on Kuruma Shrimp (Marsupenaeus japonicus)"

_microorganisms, 2022, doi:10.3390/microorganisms10020244_

Round 1

Reviewer 1 Report

This MS described the affect of Purple Non-Sulfur Bacterium, Rhodovulum sulfidophilum KKMI01, on growth of Kuruma Shrimp (Marsupenaeus japonicus). The short-term (3 days) and long-term (145 days) effects of PNSB to genes expression related to molting and innate immunity were also investigated. Overall, the MS was well written. However, there are some points that need to be addressed to make the experiments more scientifically sound.

  1. I would recommend the authors do the shrimp susceptibility test against pathogenic bacteria.
  2. It would be great if you have any data to support the expression study of immune genes (i.e. PO, SOD, lysozyme activities of hemocytes or phagocytic/clearance activity)
  3. Most statements related to genes of interest in the Results section should be moved to the Introduction section.
  4. The duration of short-term experiments (3 days) should be indicated within the Method section 2.2.
  5. The water temperature, pH, and salinity maintained during the experiments should also be mentioned in the Method section 2.2, 2.3.
  6. The replications should be specified in the RT-qPCR experiment, both biological-and technical replicates.
  7. Based on Figure 4, the significant upregulation of SOD2 was not observed in all 5 PNSB treatment conditions as mentioned in the text (Line 195).
  8. Please provide the explanation why W3 method was more effective than others?

Author Response

We greatly appreciate the reviewer’s time and effort to review our manuscript and his/her very helpful comments and suggestions. Here, we respond to the comments in the order of mention by the reviewer.

  1. I would recommend the authors do the shrimp susceptibility test against pathogenic bacteria. 

(Answer)

We totally agree with the reviewer’s suggestion that we should do the shrimp susceptibility test against pathogenic bacteria. Actually we have performed several experiments, but at present we do not have sufficient data for publication. We therefore would like to show one preliminary result as a supplementary material (Figure S2) in the revised manuscript. Please see lines 252 to 255 of the revised manuscript, and Figure S2.

  1. It would be great if you have any data to support the expression study of immune genes (i.e. PO, SOD, lysozyme activities of hemocytes or phagocytic/clearance activity)

(Answer)

We agree with reviewer’s suggestion. We, however, have not examined the activities of the enzyme related to immunity, and we would like to examine those in the future study.

  1. Most statements related to genes of interest in the Results section should be moved to the Introduction section.

(Answer)

As suggested by the reviewer, we moved the statements related to genes of interest, including proPO system, oxidative burst and antimicrobial peptides (AMPs), to the Introduction section. Please see lines 44 to 66 of the revised manuscript.

  1. The duration of short-term experiments (3 days) should be indicated within the Method section 2.2.

(Answer)

As suggested by the reviewer, we indicated the duration of short-term experiments (3 days) in the Method section 2.2 (line 84). Thank you for pointing it out.

  1. The water temperature, pH, and salinity maintained during the experiments should also be mentioned in the Method section 2.2, 2.3.

(Answer)

As suggested by the reviewer, we described the water temperature, pH, and salinity in section 2.2 (line 92 to 94 of the revised manuscript).

For section 2.3., we showed the daily record (at 15:00) of water temperature as a supplementary material (Figure S1), and described the range of pH during the experiment (pH 7.7–8.3) in the text. Please see lines 115 to 117 of the revised manuscript. However, the salinity was not shown because we have no salinity data for this experiment.

  1. The replications should be specified in the RT-qPCR experiment, both biological-and technical replicates.

(Answer)

As suggested by the reviewer, the technical replicates were shown in the Method section 2.4 as “Two technical replicate qRT-PCR reactions were run for each biological replicate (lines 143 to 144 of the revised manuscript)” The numbers of biological replicates were shown in each figure legend.

  1. Based on Figure 4, the significant upregulation of SOD2 was not observed in all 5 PNSB treatment conditions as mentioned in the text (Line 195).

(Answer)

As the reviewer pointed out, upregulation of SOD2 was not significant for DF condition. We revised the description as follows;

“the upregulation of SOD2 was observed in LF, W1, W3 and W5 conditions (Figure 3), (lines 211 to 212 of the revised manuscript)”

Thank you for pointing out our mistake.

  1. Please provide the explanation why W3 method was more effective than others?

(Answer)

As suggested by the reviewer, we discussed why W3 method was more effective than others in the Discussion section (lines 411 to 419 of the revised manuscript) as follows.

“We also examined the method of adding dead cells into the rearing water at the concentration of (103 cfu/mL equivalent), and found that this method worked as well as adding live cells (data not shown), indicating that some cellular components were responsible for the effect. The cell density of a batch culture of R. sulfidophilum KKMI01 in the stationary phase is about 109cfu/mL and 5 mg cell dry weight (d.w.)/mL. The dosage of W1 (101 cfu/mL), W3 (103 cfu/mL), and W5 (105 cfu/mL) can, therefore, be calculated to be 0.05, 5, and 500 ng cell d.w./ml, respectively. Based on this calculation, the active principle in PNSB cells, which stimulated the immunity of kuruma shrimp, was supposed to show its bioactivity at the concentration of as low as pg/ml to ng/ml order. We expect lipopolysaccharide (LPS) of PNSB as a possible candidate of this active principle, because generally LPS exhibits its bioactivity at this order [58]. In addition, Takahashi et al. reported that LPS of Pantoea agglomerans, a Gram-negative bacterium that grows symbiotically with various plants, enhanced the phagocytic activity and phenoloxidase (PO) activity in M. japonicus, and also enhance their resistance to penaeid rod shaped DNA virus (PRDV) [59]. LPS is also known as endotoxin because of its ability to induce toxic inflammatory responses. The overdosage effect of PNSB cells observed in W5 condition, may possibly be explained by the deleterious effects of LPS. The detailed study on the effects of LPS from R. sulfidophilum KKMI01 in M. japonicus are under progress in our laboratory.”

Thank you again for your quite helpful comments and suggestions.

Reviewer 2 Report

Manuscript “Probiotic Effects of a Marine Purple Non-Sulfur Bacterium, Rhodovulum sulfidophilum KKMI01, on Kuruma Shrimp 3 (Marsupenaeus japonicus)” describes bacteria induced changes in gene expression in shrimp aquaculture.  Purple non-sulfur bacteria (PNSB)  Rhodovulum sulfidophilum KKMI01 are used as probiotics. Short-term (3 days) effects of R. sulfidophilum KKMI01 on the gene expression in shrimp were examined and upregulation of several molting-related genes  as well as immune genes was shown. PNSB-treatment in 145-day outdoor experiment increased the average body weight and survival rate of shrimps. Upregulation of above-mentioned genes was also observed. KKMI01 provides a feasible probiotic candidate in shrimp aquaculture due to low effective dosage.

 Manuscript is well written, the study design is good and the methods and results are clearly represented. I only have minor suggestions for the authors:

material and methods section does not describe how long the short experiment was

rows 92-95: Please specify. It is hard to understand if the water was changed entirely after 30 days and then every week or how exactly?

Should it be mentioned in the methods part that the RNA samples were collected after 70 days in long experiment? Why this was not done at the end of the experiment?

I don’t think the figure 1 is necessary.

Author Response

We greatly appreciate the reviewer’s time and effort to review our manuscript and his/her very helpful comments and suggestions. Here, we respond to the comments in the order of mention by the reviewer.

(Comment)

material and methods section does not describe how long the short experiment was

(Answer)

As suggested by the reviewer, we indicated the duration of short-term experiments (3 days) in the Method section 2.2 (row 84 of the revised manuscript). Thank you for pointing it out.

(Comment)

rows 92-95: Please specify. It is hard to understand if the water was changed entirely after 30 days and then every week or how exactly?

(Answer)

We agree with reviewer’s suggestion that this description is not clear, and revised it as follow;

Original manuscript (rows 92-93): “The water was changed after 30 experimental days, in which approximately 10% of the water was exchanged with fresh seawater once a week,”

Revised manuscript (rows 117-119): “No water change was performed during the first 30 experimental days, after which approximately 10% of the water was exchanged with fresh seawater once a week,”

(Comment)

Should it be mentioned in the methods part that the RNA samples were collected after 70 days in long experiment? Why this was not done at the end of the experiment?

(Answer)

Actually, we isolated RNA samples from the shrimp after 70 days and 145 days (at the end of the experiment). And we partially examined the gene expression levels of some cuticle and immune proteins of shrimp sampled on day 145, but the difference between the PNSB-treated and control groups was smaller than that of shrimp sampled on day 70, so that we just described the results of 70 days. We added this explanation in the revised manuscript of “2.3. Long-term (145 days) 200-ton aquaria experiments” (rows 124-128) and “3.2. Outdoor 200-tons aquaria experiments” (rows 332-335).

(Comment)

I don’t think the figure 1 is necessary.

(Answer)

As suggested by reviewer, we removed the figure 1 in the revised manuscript.

Thank you again for your quite helpful comments and suggestions.

Round 2

Reviewer 1 Report

The revised manuscript has been significantly updated as suggested.  Only minor point in the revised manuscript is below.

Line 421, 425: Please check the fonts of references of [58], [59].